# Five-Year Trends of Vascular Disease-Related Amputations in Romania: A Retrospective Database Study

**DOI:** 10.3390/jcm13092549

**Published:** 2024-04-26

**Authors:** Horațiu F. Coman, Bogdan Stancu, Octavian A. Andercou, Razvan Ciocan, Claudia D. Gherman, Adriana Rusu, Norina A. Gavan, Cosmina I. Bondor, Alexandru D. Gavan, Cornelia G. Bala, Alexandru Necula, Trif Ana, Trif Tatiana, Peter L. Haldenwang

**Affiliations:** 1Vascular Surgery Clinic, Cluj County Emergency Hospital, 400347 Cluj-Napoca, Romania; horatiucoman@gmail.com; 2Second Department of Surgery, “Iuliu Hațieganu” University of Medicine and Pharmacy, 400006 Cluj-Napoca, Romania; bstancu7@yahoo.com; 3Department of Surgery—Practical Abilities, “Iuliu Hațieganu” University of Medicine and Pharmacy, 400337 Cluj-Napoca, Romania; razvan.alexandru.ciocan@gmail.com (R.C.); ghermanclaudia@yahoo.com (C.D.G.); 4Department of Diabetes and Nutrition Diseases, “Iuliu Hațieganu” University of Medicine and Pharmacy, 400006 Cluj-Napoca, Romania; adriana.rusu@alex-ink.net (A.R.); cbala@umfcluj.ro (C.G.B.); 5Wörwag Pharma Romania SRL, 400267 Cluj-Napoca, Romania; norina.gavan@woerwagpharma.ro; 6Department of Medical Informatics and Biostatistics, “Iuliu Hațieganu” University of Medicine and Pharmacy, 400349 Cluj-Napoca, Romania; cosmina_ioana@yahoo.com; 7Department of Medical Devices, “Iuliu Hațieganu” University of Medicine and Pharmacy, 400349 Cluj-Napoca, Romania; gavan.alexandru@umfcluj.ro; 8Faculty of Medicine, “Iuliu Hațieganu” University of Medicine and Pharmacy, 400349 Cluj-Napoca, Romania; alexnecula10@gmail.com; 9Vascular Surgery Department, “Nicolae Stăncioiu” Heart Institute, 400001 Cluj-Napoca, Romania; trif.t.a.ana@gmail.com; 10Anesthesia and Intensive Care Department, Regional Institute of Gastroenterology & Hepatology “Prof. Dr. Octavian Fodor”, 400394 Cluj-Napoca, Romania; trif_t_tatiana@yahoo.com; 11Department for Cardiothoracic Surgery, University Hospital Bergmannsheil Bochum, Ruhr-University of Bochum, 44789 Bochum, Germany; peter.haldenwang@bergmannsheil.de

**Keywords:** lower extremity amputations, peripheral artery disease, arterial embolism and thrombosis, retrospective study

## Abstract

**Background/Objectives:** Lower extremity amputations (LEAs) are a burdensome complication of peripheral artery disease (PAD) and/or arterial embolism and thrombosis (AET). We assessed the trends in PAD- and/or AET-related LEAs in Romania. **Methods:** This retrospective study (2015–2019) analyzed data on minor and major LEAs in hospitalized patients recorded in the National School for Public Health, Management, and Health Education database. The absolute numbers and incidences of LEAs were analyzed by diagnosis type, year, age, sex, and amputation level. **Results:** Of 38,590 vascular disease-related amputations recorded nationwide, 36,162 were in PAD and 2428 in AET patients. The average LEA incidence in the general population was 34.73 (minimum: 31.96 in 2015; maximum: 36.57 in 2019). The average incidence of major amputations, amputations above the knee, hip amputations, amputations below the knee, and minor amputations was 16.21 (15.62 in 2015; 16.84 in 2018), 13.76 (13.33 in 2015; 14.28 in 2018), 0.29 (0.22 in 2017; 0.35 in 2019), 2.15 (2.00 in 2015; 2.28 in 2019), and 18.52 (16.34 in 2015; 20.12 in 2019), respectively. Yearly PAD- and/or AET-related amputations were significantly higher in men versus women. The overall number of LEAs increased with age, particularly in patients ≥ 70 years. The increase in the total number of amputations was mainly due to a constant rise in minor amputations for both groups, regardless of gender. **Conclusions**: PAD- and/or AET-related LEAs in Romania increased from 2015 to 2019, with men having a greater incidence than women. Raising awareness and effective management strategies are needed to prevent LEAs.

## 1. Introduction

Atherosclerosis and its complications, such as peripheral artery disease (PAD) and arterial embolism and thrombosis (AET), represent a considerable overload for both the patients and the healthcare system. From 2015 to 2019, a concerning increase in the number of lower extremity amputations (LEAs) has been observed across many European countries, primarily driven by the rising prevalence of PAD and other contributing comorbidities. Nowadays, PAD poses a significant global health challenge due to lifestyle factors and associated comorbidities. While the prevalence of PAD continues to increase and is often higher among men and the elderly population, women from lower socioeconomic backgrounds experience a higher incidence, with rates twice as high as those recorded in men [1].

A narrative review published in 2020 highlighted a substantial disparity in the burden of cardiovascular diseases between Western European countries (WECs) and Central and Eastern European countries (CEECs) [2]. The CEECs comprise Bulgaria, Croatia, the Czech Republic, Estonia, Latvia, Lithuania, Hungary, Poland, Romania, Slovenia, and the Slovak Republic. Considering that these nations represent approximately 20% of the entire European Union population, it emphasizes the importance of this health issue [3]. Evaluation of the rates of major LEAs from each country provides insight into the performance of the entire healthcare system, as the incidence of these amputations serves as a central outcome indicator [4]. It is important to note that, despite significant advances in international strategies, as well as in novel techniques and vascular devices for limb salvage [5,6,7], there are discrepancies between the healthcare systems of WECs and CEECs.

A comprehensive review examined disparities in vascular risk factors, healthcare structures, processes, and outcomes in peripheral vascular care between WECs and CEECs, highlighting that CEECs face challenges in public health expenditure, access to healthcare services, and availability of medical professionals compared to their Western counterparts. Analysis of healthcare processes revealed disparities in the volume and timing of vascular procedures, with CEECs lagging behind WECs in adopting endovascular-first strategies and achieving optimal cholesterol levels among patients. Additionally, outcomes such as cardiovascular disease mortality and lower limb amputation rates underscore the significant gap between the two regions, with CEECs experiencing higher mortality rates and a greater incidence of major amputations. Moreover, this review highlighted the limited availability of robust scientific data from CEECs, which impedes meaningful comparisons and hinders evidence-based policymaking. Addressing this data gap is crucial for understanding and addressing disparities in vascular care across Europe [8].

Therefore, there is an imperative need to raise awareness about this significant healthcare problem and to drive clinical practice actions that reduce amputation rates in patients with PAD and/or AET and, consecutively, the burden on healthcare systems. 

In this retrospective database study, we assessed the rates of LEAs related to PAD and/or AET by age, sex, and amputation level in Romania during the 2015–2019 timeframe.

## 2. Materials and Methods

We conducted a retrospective analysis of PAD- and/or AET-related LEAs in patients admitted to public hospitals in Romania between 1 January 2015 and 31 December 2019.

The data were retrieved from the National School for Public Health, Management, and Health Education (Școala Națională de Sănătate Publică, Management și Perfecționare). This database collects all hospital admissions in Romania [9]. The following variables were analyzed in our study: the total number and level of amputation, the type of vascular disease (PAD and/or AET) that caused amputation, and the age and gender of patients who underwent amputations. Amputation procedures were considered major if LEA occurred above the ankle or at ankle level through the malleoli of the tibia and fibula and minor if LEA included ankle disarticulation or occurred below the ankle; traumatic LEAs were excluded. PAD included any medical condition that caused reduced blood flow to the lower limbs. AET included any medical event leading to embolism and thrombosis of arteries of the lower extremities. 

The following criteria were used to collect relevant data:A confirmed diagnosis of primary or secondary PAD and/or AET at discharge (International Classification of Diseases 10 [ICD 10] codes I70.0–I70.9, I74.0–I74.9).The patient underwent a major amputation procedure during the study period including the following types of amputations: hip (ICD 10 AM v.3 code 44370-00), above or below the knee (codes 44367-00, 44367-02), and ankle through the malleoli of the tibia and fibula (code 44361-01).The patient underwent a minor amputation procedure during the study period including ankle disarticulation (44361-00), mid-tarsal and trans-metatarsal amputation (codes 44364-00, 44364-01), toe amputations with or without metatarsal bone (codes 44358-00, 44338-00), and toe disarticulation (code 90557-00).

Descriptive analysis was used to provide an overall overview of clinical practices and to summarize the data. To estimate the crude incidence of amputations caused by vascular diseases/100,000 persons/year, we analyzed the population data provided by the National Institute of Statistics (to ascertain the official population of Romania between 2015 and 2019) [10,11,12,13]. To estimate incidences by year in the general population and patients with PAD and/or AET of lower extremity arteries, the yearly data of the whole population were applied. To estimate the incidence rates stratified by sex, type of disease, and year, specific population size data were applied. The rates of minor to major amputations, as well as femoral to crural major amputations, were also calculated.

The statistical analysis was carried out using Microsoft Excel, Version 2301 (Build 16026.20200 Click-to-Run), Microsoft Corporation, Redmond, WA, USA. The trend analysis was performed using the Chi-square test. The level of statistical significance was set at 0.05 and the confidence level at 95%.

## 3. Results

From 2015 to 2019, a total number of 38,590 PAD- and/or AET-related amputations) were recorded nationwide. Of these, 36,162 (93.70%) were recorded in patients with PAD and 2428 (6.30%) in patients with AET of lower extremity arteries. The distribution of LEA per 100,000 persons/year by province in Romania is depicted in Figure 1.

Table 1 presents a detailed yearly list of the absolute number of vascular disease-related amputations of lower extremities in the general population and the occurrence rate by amputation type and level. Crude incidences corresponding to each category are presented in Table 2.

Throughout the study period, the mean (±standard deviation [SD]) number of amputations per year was 7718 (±381.42). According to the vascular disease type, the mean (±SD) numbers of LEAs were 7232.4 (±374.97) in patients with PAD, and 485.6 (±16.82) in patients with AET of lower extremity arteries.

An increase in the number of amputations among patients with PAD was observed. Compared to 2015, the number of amputations increased by 7.26%, 9.28%, 12.76%, and 14.40% in 2016, 2017, 2018, and 2019, respectively. In patients with AET of lower extremity arteries, the number of amputations increased by 5.46% in 2016, decreased by 2.94% in 2017, and increased by 2.52% in 2018 and 5.04% in 2019 compared to 2015 (Figure 2).

In patients with PAD, the absolute number of major amputations fluctuated between 2015 and 2019, peaking in 2018, and decreasing in 2019. For minor amputations, the absolute number increased annually, demonstrating a steady upward trajectory (Figure 3).

In patients with AET of lower extremity arteries, the absolute number of major amputations fluctuated during the study period, with the highest number recorded in 2015, followed by a decreasing trend until 2017. Another peak was observed in 2018, followed by a slight decrease in 2019. The absolute number of minor amputations tended to increase until 2016, followed by a decreasing trend until 2018, and a rise in 2019 (Figure 4).

The yearly minor–major amputation ratio was higher in patients with AET of lower extremity arteries compared to those with PAD: 1.16 vs. 1.04 in 2015, 1.64 vs. 1.10 in 2016, 1.72 vs. 1.15 in 2017, 1.38 vs. 1.12 in 2018, and 1.58 vs. 1.20 in 2019.

Overall, women with PAD and AET had a lower number of amputations compared to men (Figure 5). Changes in the amputation numbers relative to the baseline (2015) showed an increasing trend of 12.95%, 8.64%, 14.14%, and 23.47% in 2016, 2017, 2018, and 2019, respectively in women with PAD. Similarly, for men, an increasing trend of 5.46%, 9.48%, 12.33%, and 11.54%, respectively, was observed. 

In contrast, among patients with AET of lower extremity arteries, there was an upward trend for women in 2016 (2.5%), followed by a decrease in 2017 (−10.83%), a slight increase in 2018 (0.83%), and another decrease in 2019 (−2.5%) compared to the baseline (2015). Similarly, for men, there was an increase in 2016 (6.46%), followed by a decrease in 2017 (−0.28%), another increase in 2018 (3.09%), and a further increase in 2019 (7.58%) compared to the baseline (2015).

Table 3 presents the crude incidence of amputations in the general population by vascular disease and sex. The yearly men–women ratio of overall amputations was 3.16, 2.95, 3.19, 3.11, and 2.86 in patients with PAD, with a median of 3.11, whereas in patients with AET of lower extremity arteries, this ratio was 2.97, 3.08, 3.32, 3.03, and 3.27, with a median of 3.08.

The incidence of major amputations was lower in women with PAD vs. men across all study years: 0.33 in 2015, 0.35 in 2016, 0.35 in 2017, 0.33 in 2018, and 0.35 in 2019. Similarly, women with AET of lower extremity arteries faced a higher risk of major amputations vs. men across all study years: 0.41 in 2015, 0.30 in 2016, 0.36 in 2017, 0.33 in 2018, and 0.31 in 2019.

The ratio of minor to major amputations was consistently lower in women with PAD vs. men: 0.918 vs. 1.079 in 2015, 0.986 vs. 1.137 in 2016, 1.047 vs. 1.188 in 2017, 1.022 vs. 1.157 in 2018, and 1.110 vs. 1.237 in 2019. Similarly, for AET of lower extremity arteries, women had a lower ratio than men in most years: 0.818 vs. 1.312 in 2015, 1.488 vs. 1.795 in 2017, 1.283 vs. 1.414 in 2018, and 1.489 vs. 1.605 in 2019, except for 2016, when the ratio of minor to major was lower in men vs. women (1.674 vs. 1.632, respectively). 

For women with PAD, the incidence significantly increased by 0.88 (*p*-value of 0.0002) from 2015 to 2016, followed by a decrease of 1.04 (*p*-value of 0.2573) from 2016 to 2017. In the subsequent years, the incidence increased by 0.95 (*p*-value of 0.1286) from 2017 to 2018 and by 0.92 (*p*-value of 0.0139) from 2018 to 2019 (Figure 6).

For men with PAD, the incidence increased by 0.95 (*p*-value of 0.0045) from 2015 to 2016, by 0.96 (*p*-value of 0.0467) from 2016 to 2017, by 0.97 (*p*-value of 0.1558) from 2017 to 2018, and decreased by 1.01 (*p*-value of 0.7361) from 2018 to 2019 (Figure 7).

For women with AET, the incidence increased by 0.97 (*p*-value of 0.8319) from 2015 to 2016, followed by a decrease of 1.15 (*p*-value of 0.2945) from 2016 to 2017. From 2017 to 2018, the incidence increased by 0.88 (*p*-value of 0.3480) and then decreased by 1.03 (*p*-value of 0.8032) from 2018 to 2019 (Figure 8).

For men with AET, the incidence increased by 0.94 (*p*-value of 0.3772) from 2015 to 2016, decreased by 1.07 (*p*-value of 0.3813) from 2016 to 2017, increased by 0.97 (*p*-value of 0.6444) from 2017 to 2018, and increased with 0.96 (*p*-value of 0.5522) from 2018 to 2019 (Figure 9).

Among patients with PAD, the number of LEAs increased more prominently in women than in men, as depicted in Figure 10. Compared to 2015, the percentage change in the number of major amputations in women vs. men with PAD was as follows: 9.12% vs. 2.63% in 2016, 1.80% vs. 4.03% in 2017, 8.28% vs. 8.27% in 2018, and 12.24% vs. 3.70% in 2019. The variations in the number of minor amputations were 17.12% vs. 8.08% in 2016, 16.08% vs. 14.53% in 2017, 20.52% vs. 16.09% in 2018, and 35.69% vs. 18.80% in 2019.

Among patients with AET of LEAs, women had a more pronounced increase in the number of minor amputations compared to men and a decrease in the number of major amputations (Figure 11). Compared to 2015, the percentage change in the number of minor amputations in women vs. men with AET was as follows: 42.59% vs. 16.34% in 2016, 18.52% vs. 12.87% in 2017, 25.93% vs. 6.44% in 2018, and 29.63% vs. 16.83% in 2019. The variations in the number of major amputations (women vs. men) were −30.30% vs. −6.49% in 2016, −34.85% vs. −17.53% in 2017, −19.70% vs. −1.30% in 2018, and −28.79% vs. −4.55% in 2019.

For PAD, the incidence of LEAs was higher for men than women by 0.30 (*p*-value of 0.0000) in 2015, 0.32 (*p*-value of 0.0000) in 2016, 0.30 (*p*-value of 0.0000) in 2017, 0.31 (*p*-value of 0.0000) in 2018, and 0.33 (*p*-value of 0.0000) in 2019 (Figure 12).

For AET, the incidence of LEAs was higher for men than women: 0.32 (*p*-value of 0.0000) in 2015, 0.31 (*p*-value of 0.0000) in 2016, 0.29 (*p*-value of 0.0000) in 2017, 0.31 (*p*-value of 0.0000) in 2018, and 0.29 (*p*-value of 0.0000) in 2019 (Figure 13).

In patients with PAD, the incidence of amputations increased with age (Figure 14). A similar trend was observed in patients with AET of LEAs (Figure 15).

In both the PAD and AET groups, the incidence of amputations increased between 2015 and 2019 and showed an age-dependent pattern. The lowest incidence was recorded in the <30 age group, and the highest incidence was in the ≥70 years age group, demonstrating an increasing trend with age. 

Furthermore, among patients with PAD aged ≥ 70 years, the incidence of amputations showed a consistent increase over the study period, reaching a peak in 2019. In contrast, in patients with AET of the same age group, the incidence of amputation showed a decreasing trend until 2017, followed by an increasing trend with a peak in 2019.

All these trends are summarized in Table 4, Table 5 and Table 6.

## 4. Discussion

Our retrospective database study evaluates PAD- and/or AET-related amputations in Romania from 2015 to 2019, aiming to provide insight into the future burden on the healthcare system.

Overall, a total of 38,590 vascular disease-related amputations (PAD and/or AET) were recorded nationwide, 36,162 of which were in persons with PAD, and 2428 in patients with AET of lower extremity arteries. The incidence of amputations in men was higher than in women for all studied years, regardless of the cause (PAD- and/or AET-related). In women, the incidence of PAD-caused amputations was higher in 2015 compared to 2016, and in 2018 compared to 2019. In men, the incidence of AET-caused amputations was higher in 2015 compared to 2016 and in 2016 compared to 2017. This increase was due to an increase in both major and minor amputations, especially in older individuals. The increasing tendency of minor-to-major amputations observed in Romania during the study period showed a trend similar to other European countries. For example, in Germany, it has been reported a decreasing trend in major LEA and an ascending trend in minor amputations during a similar timeframe (2015–2019) [1]. A study conducted in Hungary from 2004 to 2017 revealed that the number of minor amputations decreased moderately after 2015, while major amputations declined slightly (15%) starting from 2013 and had a more marked decrease (22%) after adjusting for age. This study shows a 10-year lag when compared to Western European countries and the US [15].

In Romania, the number of vascular specialists has increased over the last few years, thus increasing the addressability of at-risk patients. Sadly, there is an unequal distribution of vascular specialists in different Romanian hospitals in different regions and this still leads to a high number of amputations before any type of vascular consultation. Due to the multi-societal efforts of the Romanian Society of Vascular Surgery, Romanian Society of Diabetes, and Romanian Society of Diabetic Neuropathy, and a slight improvement in funding, there has been a more holistic approach of these patients in recent years, which positively influenced the limb salvage strategies. With the improvement of cooperation between different specialties, multidisciplinary teams are evolving, and, thus, more physicians can perform vascular procedures, reducing the timeframe between symptoms’ onset and medical care.

If we analyze the data from a demographical perspective, Romania reflects a similar pattern with the rest of the European countries, with older male patients aged 60–69 and >70 years exhibiting the highest prevalence of LEA due to PAD and AET. While the incidence among younger patients remains lower compared to older age groups, the significant numbers observed in the <50-year-old population could potentially create a shift in disease demographics in the coming years for the young population, warranting proactive intervention measures.

Gender disparities were also evident over the study period. For AET, a decrease in major amputations was observed in women compared to men, which contrasts with the increase observed for PAD. The underlying reasons for this difference remain unclear, with possible explanations including variations in healthcare-seeking behaviors and psychological responses to chronic diseases across different genders.

Variations in amputation rates have been documented within and between countries, including Denmark and other European countries [16,17,18,19]. Still, the rise in amputation rates in the Central Region of Denmark caused a public scandal that thrust vascular surgery into the spotlight of Danish public attention. A study published in 2023 found that a population with a high prevalence of atherosclerosis may have a high number of vascular procedures and amputations, while a fit and healthy population may have low numbers of both. However, in the Central Region of Denmark, there were many amputations reported but few vascular procedures, indicating a concerning imbalance [20]. In our study, the regions with the highest amputation rates are also the regions that provide the most advanced medical and vascular care in their regional hospitals and therefore treat more patients annually. Also, due to the lack of adequate vascular care in some regions, the patients were referred to nearby, bigger, regional public hospitals, artificially increasing the amputation rates in these regions.

This practice was also reported in other studies, with one notable observation being the correlation between the level of regional development, the quality of healthcare, and the high number of amputations. Factors such as restricted healthcare access and delays in hospital presentation contribute to unfortunate outcomes for patients, particularly in areas with limited vascular and endovascular care options. Endovascular treatment has shown lower morbidity rates in PAD patients where available [21].

One positive trend was seen in the decreasing number of major amputations among patients with AET. This trend may imply (at least theoretically) that these patients, due to the onset of acute symptoms, may potentially benefit from quicker access to limb salvage treatments and have improved outcomes. In comparison to the major LEA, the number of minor amputations in patients with AET showed an increase, mirroring the trend observed in patients with PAD.

Our study has some limitations arising from the retrospective nature of data collection. The analysis only included cases of LEAs where a patient underwent a single amputation, as we lacked information on multiple amputations in one person. Unfortunately, follow-up information on amputation outcomes and long-term mortality rates was not available.

Due to the coding system and the way the database was structured, we cannot make any relevant statements as to whether PAD was the main cause of the amputation or just a secondary diagnosis coded at discharge. Even though this is a limitation, our study still underscores the increasing trend of amputation rates in patients with PAD. In contrast, this was not observed in patients with a clear first diagnosis of AET at discharge. The same limitation was encountered in a German study [22].

Another limitation is that the data obtained are procedure-related, not patient-related. Subsequently, it is not possible to establish a valid connection between the absolute number of amputations and the absolute number of patients. Therefore, one patient may have had more than one amputation and more than one limb affected. In addition, these data do not distinguish whether a patient has multiple amputations in a year or across multiple years. The same individual may have been repeatedly counted as separate cases annually or over several years, which can inflate the perceived frequency of the condition and impact the accuracy of this study’s conclusions regarding the true incidence of amputations in Romania.

There is a lack of information on the patient- and healthcare-related factors, such as diabetes duration, reasons for amputations, ethnic background, social status, and clinical or laboratory parameters. These limitations hindered our ability to investigate the impact of limb salvage interventions, such as debridement or revascularization procedures, on the risk of amputation. Regarding the relationship between amputations and vascular procedures, a study conducted in Denmark suggested that a high number of vascular procedures may coexist with a high number of amputations in a population with a high prevalence of atherosclerosis [20]. Likewise, a low number of vascular procedures may coexist with a low number of amputations in a fit and healthy population.

A future prospective study, designed to address the complex issues related to amputations, including the understanding of their intricate nature, should help mitigate these limitations [23].

Although this study did not aim to address preventive strategies in lowering the LEAs in PAD and AET patients, we believe that improving the screening process, addressability, and continued medical education for both healthcare providers and patients will result in lowering the amputation rates and the social impact of this burden.

## 5. Conclusions

PAD- and AET-related lower extremity amputations in Romania increased from 2015 to 2019, with men having a greater incidence than women. The increase in the ratio of minor–major amputation was higher in patients with AET of lower extremity arteries than those with PAD, with both groups showing an upward trend in minor amputations.

Age did play a role in influencing the amputation rates for both PAD and/or AET patient groups, with the number of amputations showing an increasing trend with age.

This situation requires immediate attention, prompting comprehensive investigations to understand its complexities and develop effective management strategies focused on preventing lower extremity amputations among patients with PAD and/or AET. Raising awareness among healthcare providers nationwide could improve management strategies and patient outcomes.

## Figures and Tables

**Figure 1 jcm-13-02549-f001:**
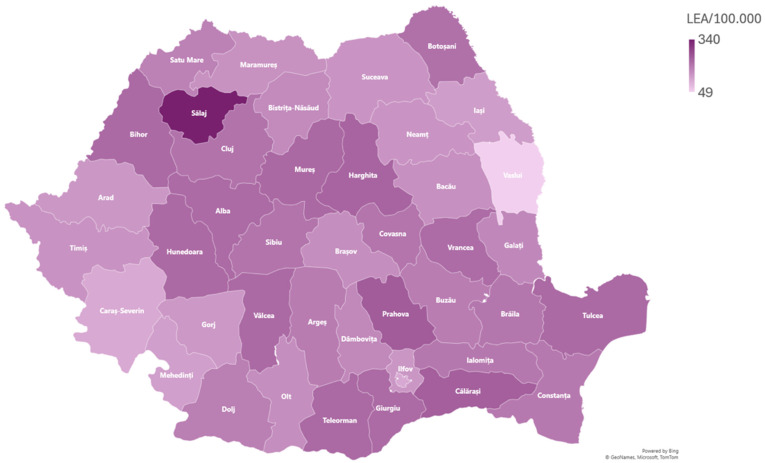
Map representation of LEAs per 100,000 population by province in Romania (2015–2019). LEA, lower extremity amputation.

**Figure 2 jcm-13-02549-f002:**
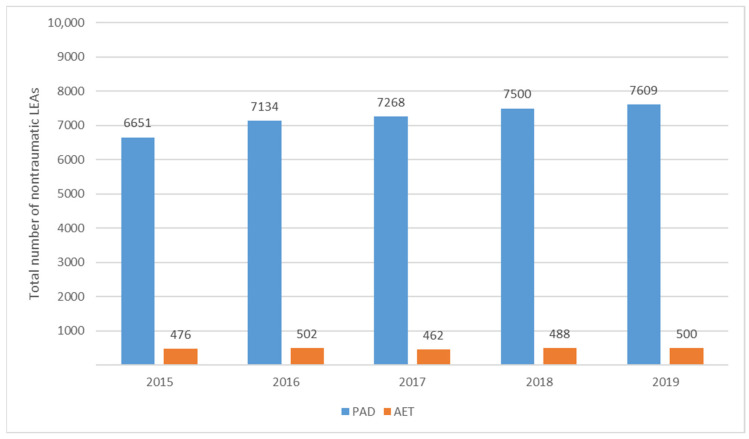
The total number of nontraumatic LEAs per year by type of vascular disease. LEAs, lower extremity amputations per year; PAD, peripheral arterial disease; AET, arterial embolism and thrombosis of lower extremity arteries.

**Figure 3 jcm-13-02549-f003:**
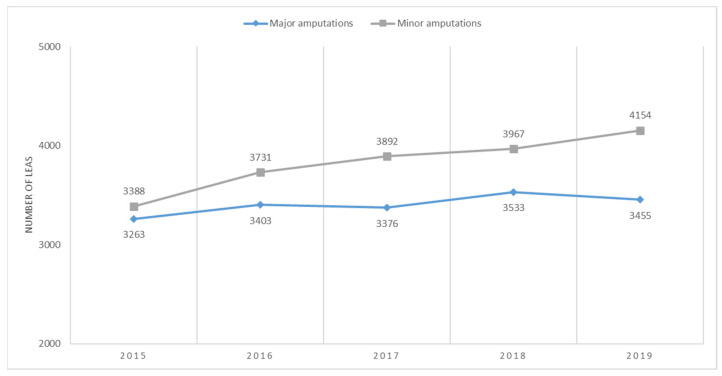
Trends in the occurrence of major and minor LEAs in patients with PAD. PAD, peripheral arterial disease.

**Figure 4 jcm-13-02549-f004:**
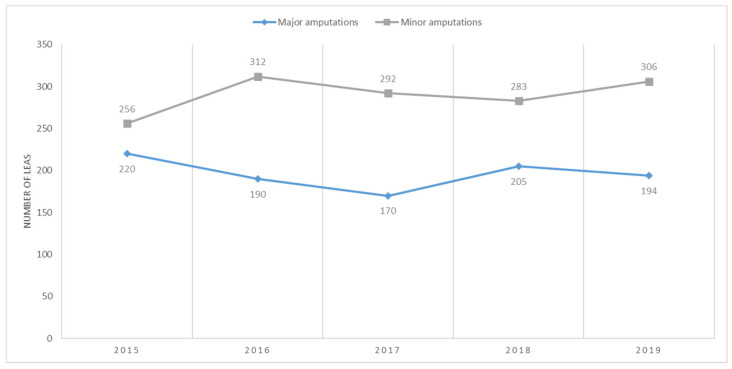
Trends in the occurrence of major and minor LEAs in patients with AET. LEAs, lower extremity amputations per year; AET, arterial embolism and thrombosis of lower extremity arteries.

**Figure 5 jcm-13-02549-f005:**
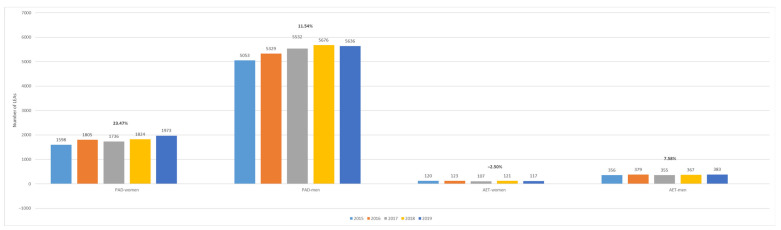
Trends in the occurrence of LEAs in patients with PAD and AET by sex. LEAs, lower extremity amputations; PAD, peripheral arterial disease; AET, arterial embolism and thrombosis of lower extremity arteries.

**Figure 6 jcm-13-02549-f006:**
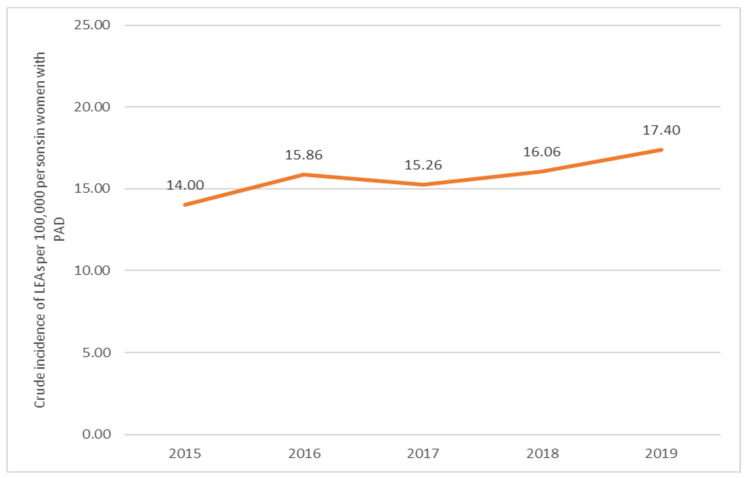
Trends in the occurrence of LEAs in women with PAD (2015–2019). PAD, peripheral arterial disease.

**Figure 7 jcm-13-02549-f007:**
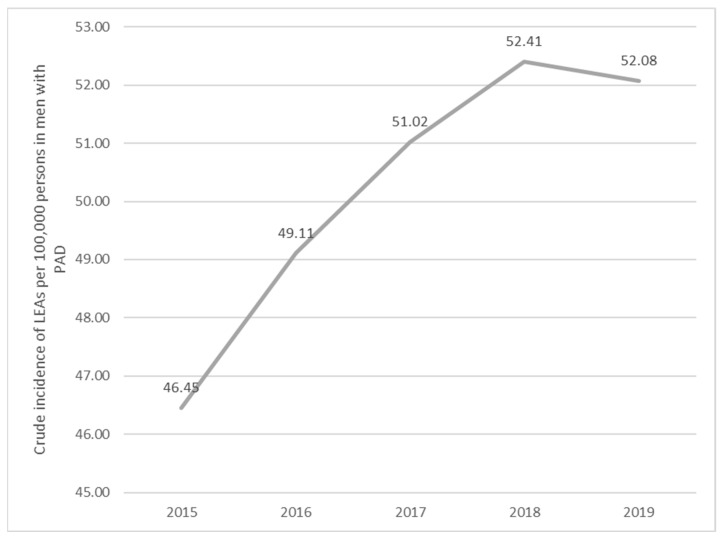
Trends in the occurrence of LEAs in men with PAD (2015–2019). PAD, peripheral arterial disease.

**Figure 8 jcm-13-02549-f008:**
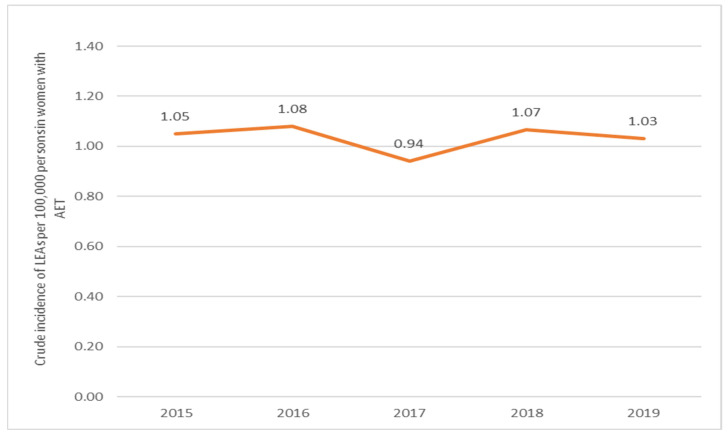
Trends in the occurrence of LEAs in women with AET (2015–2019). AET, arterial embolism and thrombosis of lower extremity arteries.

**Figure 9 jcm-13-02549-f009:**
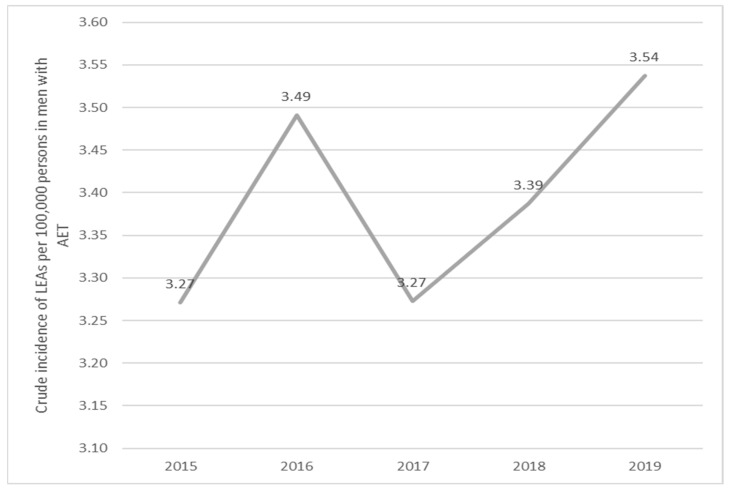
Trends in the occurrence of LEAs in men with AET (2015–2019). AET, arterial embolism and thrombosis of lower extremity arteries.

**Figure 10 jcm-13-02549-f010:**
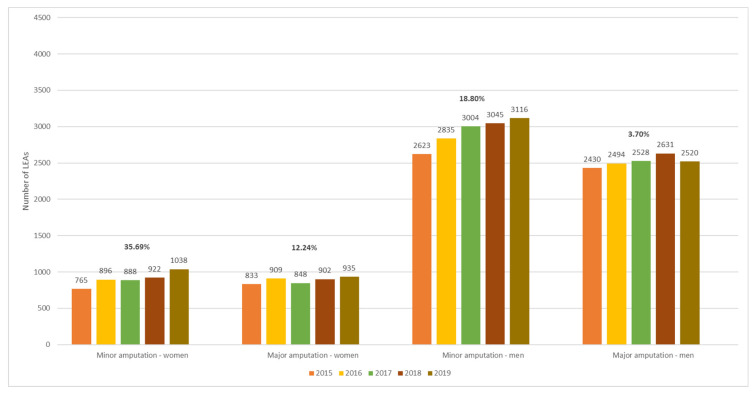
Trends in the occurrence of LEAs in patients with PAD by sex and level of amputation. LEAs, lower extremity amputations; PAD, peripheral arterial disease.

**Figure 11 jcm-13-02549-f011:**
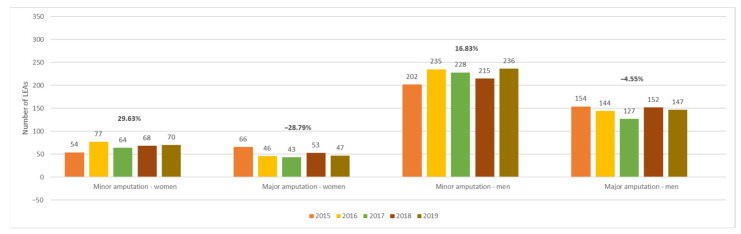
Trends in the occurrence of LEAs in patients with AET by sex and level of amputation. LEAs, lower extremity amputations; AET, arterial embolism and thrombosis of lower extremity arteries.

**Figure 12 jcm-13-02549-f012:**
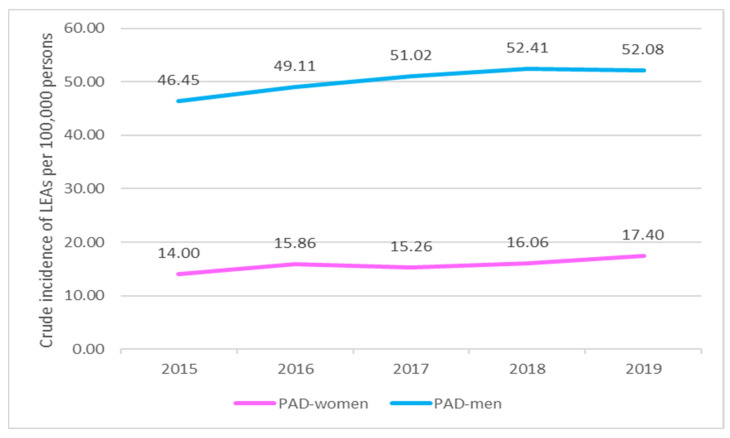
Comparison of the trends in the occurrence of LEAs between men and women with PAD (2015–2019). LEAs, lower extremity amputations; PAD, peripheral arterial disease.

**Figure 13 jcm-13-02549-f013:**
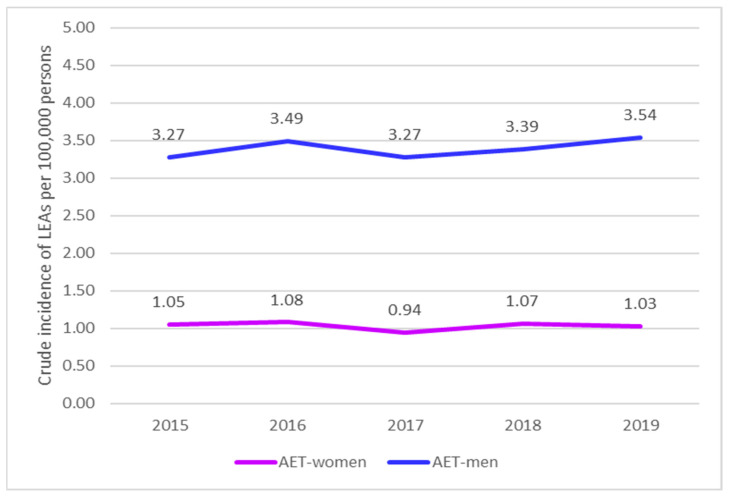
Comparison of the trends in the occurrence of LEAs between men and women with AET (2015–2019); LEAs, lower extremity amputations; AET, arterial embolism and thrombosis of lower extremity arteries.

**Figure 14 jcm-13-02549-f014:**
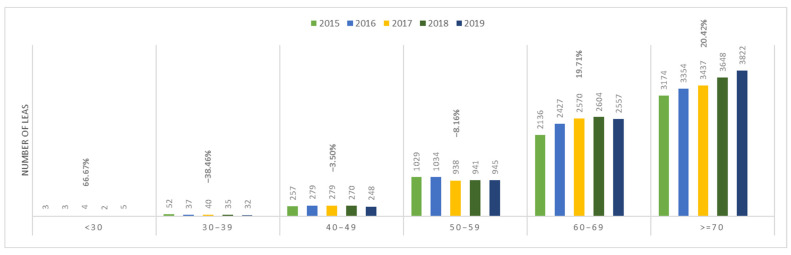
Trends in the occurrence of LEAs in patients with PAD by age. LEAs, lower extremity amputations; PAD, peripheral arterial disease.

**Figure 15 jcm-13-02549-f015:**
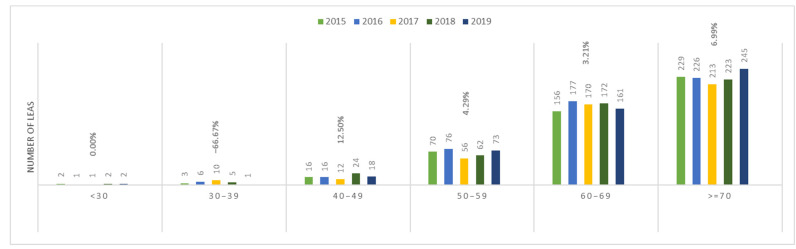
Trends in the occurrence of lower LEAs in patients with AET by age. LEAs, lower extremity amputations; AET, arterial embolism and thrombosis of lower extremity arteries.

**Table 1 jcm-13-02549-t001:** Yearly absolute number and occurrence rates of LEAs by amputation type and level.

	2015 [10]	2016 [10]	2017 [11]	2018 [12]	2019 [13]	Total
Amputation procedures, No.	7127	7636	7730	7988	8109	38,590
Major amputations (% of total amputations)	3483 (48.9%)	3593 (47.1%)	3546 (45.9%)	3738 (46.8%)	3649 (45.0%)	18,009 (46.7%)
Amputations above the knee (% of total amputations)	2972 (85.33%)	3070 (85.44%)	3013 (84.97%)	3170 (84.80%)	3067 (84.05%)	15,292 (84.91%)
Hip amputations (% of total amputations)	66 (1.89%)	64 (1.78%)	48 (1.35%)	72 (1.93%)	77 (2.11%)	327 (1.82%)
Amputation below the knee (% of total amputations)	445 (12.78%)	459 (12.77%)	485 (13.68%)	496 (13.27%)	505 (13.84%)	2390 (13.27%)
Minor amputations (% of total amputations)	3644 (51.1%)	4043 (52.9%)	4184 (54.1%)	4250 (53.2%)	4460 (55.0%)	20,581 (53.3%)

Data on Romania’s population for each study year were collected from the National Institute of Statistics database (references included in the table header). Data on the population with vascular diseases for each study year were collected from the National Institute of Public Health database (reference included in the corresponding row); LEAs, lower extremity amputations; No., absolute number.

**Table 2 jcm-13-02549-t002:** Yearly crude incidences of LEAs in the general population, and by amputation type and level.

Incidence (/100,000 Persons/Year)	2015 [10]	2016 [10]	2017 [11]	2018 [12]	2019 [13]	Average Values
Vascular diseases-related amputations in the general population [14]	31.96	34.33	34.78	35.99	36.57	34.73
Major amputations	15.62	16.15	15.96	16.84	16.46	16.21
Amputations above the knee	13.33	13.80	13.56	14.28	13.83	13.76
Hip amputations	0.30	0.29	0.22	0.32	0.35	0.29
Amputation below the knee	2.00	2.06	2.18	2.23	2.28	2.15
Minor amputations	16.34	18.18	18.83	19.15	20.12	18.52

Data on Romania’s population for each study year were collected from the National Institute of Statistics database (references included in the table header). Data on the population with vascular diseases for each study year were collected from the National Institute of Public Health database (reference included in the corresponding row); LEAs, lower extremity amputations.

**Table 3 jcm-13-02549-t003:** Crude incidence by vascular disease and sex.

	2015 [10]	2016 [10]	2017 [11]	2018 [12]	2019 [13]	Total
Men–women ratio of risk of amputations in the general population	3.30	3.10	3.35	3.26	3.02	3.20 *
Incidence of amputations caused by vascular diseases in women/100,000 women/year	15.05	16.93	16.20	17.12	18.43	NA
Incidence of amputations caused by vascular diseases in men/100,000 men/year	49.70	52.58	54.27	55.77	55.59	NA

Data on Romania’s population for each study year were collected from the National Institute of Statistics database (references included in the table header). The incidence of vascular disease-related amputations was calculated using the sex-specific population size. * Risk was computed using the average general population size over the 5-year study period for both women and men.

**Table 4 jcm-13-02549-t004:** Incidence of vascular diseases (both PAD and AET) to LEAs/100,000 persons/year by age in the general population.

Age Group	2015 [10]	2016 [10]	2017 [11]	2018 [12]	2019 [13]
<30	0.07	0.05	0.07	0.06	0.10
30–39	1.53	1.21	1.41	1.13	0.94
40–49	7.67	7.99	7.60	7.81	7.20
50–59	39.57	41.48	38.50	36.99	35.74
60–69	95.08	102.91	105.33	104.19	100.69
>70	142.68	151.46	153.22	161.71	167.90

Data on Romania’s population for each study year were collected from the National Institute of Statistics database (references included in the table). PAD, peripheral arterial disease; AET, arterial embolism and thrombosis of lower extremity arteries; LEAs, lower extremity amputations.

**Table 5 jcm-13-02549-t005:** Incidence of PAD to LEAs/100,000 persons/year by age in the general population.

Age Group	2015 [10]	2016 [10]	2017 [11]	2018 [12]	2019 [13]
<30	0.04	0.04	0.05	0.03	0.07
30–39	1.44	1.04	1.13	0.99	0.91
40–49	7.22	7.56	7.29	7.18	6.71
50–59	37.05	38.64	36.33	34.70	33.18
60–69	88.61	95.91	98.79	97.73	94.73
>70	133.08	141.90	144.28	152.39	157.78

Data on Romania’s population for each study year were collected from the National Institute of Statistics database (references included in the table). PAD, peripheral arterial disease; LEAs, lower extremity amputations.

**Table 6 jcm-13-02549-t006:** Incidence of AET to LEAs/100,000 persons/year by age in the general population.

Age Group	2015 [10]	2016 [10]	2017 [11]	2018 [12]	2019 [13]
<30	0.03	0.01	0.01	0.03	0.03
30–39	0.08	0.17	0.28	0.14	0.03
40–49	0.45	0.43	0.31	0.64	0.49
50–59	2.52	2.84	2.17	2.29	2.56
60–69	6.47	6.99	6.53	6.46	5.96
>70	9.60	9.56	8.94	9.32	10.11

Data on Romania’s population for each study year were collected from the National Institute of Statistics database (references included in the table for each year). AET, arterial embolism and thrombosis of lower extremity arteries.

## Data Availability

Restrictions apply to the availability of the data obtained from the National School for Public Health, Management, and Health Education. These data are available from the authors with the permission of the National School for Public Health, Management, and Health Education.

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
