# Peer review of "Five-Year Trends of Vascular Disease-Related Amputations in Romania: A Retrospective Database Study"

_jcm, 2024, doi:10.3390/jcm13092549_

Round 1

Reviewer 1 Report

Comments and Suggestions for Authors

It is an important paper from CEECs about amputation practice in Romania. My concerns and comments are summarized below.

 Introduction

Line 62-74

Comment (1) this part highlighting the importance of differences between CEECs and WECs in many aspects is a reasonable positioning, however, I would like to recommend incorporating a review, explicitly addressing the differences in vascular diseases, published in JCM (Kolossváry E, Björck M, Behrendt CA. A Divide between the Western European and the Central and Eastern European Countries in the Peripheral Vascular Field: A Narrative Review of the Literature. J Clin Med. 2021 Aug 12;10(16):3553.)

Methods

Line 81-113

Comment (2) The term of ’PAD and AET’ is not consequent. I think ’PAD and/or AET’ would be more appropriate.

Comment (3) I am just wondering whether other ICD codes could have been available. Indication of the presence of diabetes is broadly expected when it is about amputations.

Line 94-98

Comment (4) An important indicator is the femoral to crural major amputations. The procedure codes that were used would make it possible to give these numbers. The authors would not consider providing these data?

Line 103-105

„To estimate the crude incidence of amputations related to vascular 103 diseases /100,000 persons/year for both the general population and patients with PAD and  AET of lower extremity arteries, we used the population data provided by the National 105 Institute of Statistics.”

Comment (5) I think for the base of crude incidences, the yearly data of whole population size was applied. In this sense, „for both the general population and patients with PAD and  AET of lower extremity arteries” is hard to understand. Additionally, when sex-stratified incidence rates are given, the base was the sex-specific population size, was not? Please clarify.

Comment (6) Please consider applying a statistical approach, even basic techniques to prove changes in trends.  

Results

Line 109-271

Comment (7) Absolute numbers of procedures and crude incidence rates are provided simultaneously but not consequently. However, absolute numbers do not help the reader to have a rough comparison with other datasets. I would prefer to focus on crude incidences alone. For example, I missed the incidence rates of major amputations, minor amputations in the whole sample and also following a stratification by sex or cause. This issue affects the tables as well. I don't know whether any advantage in offering absolute numbers exists? In this short period of observation (5 years), the Romanian population (population at risk) decreased by more than 2%. Consequently, the absolute numbers are misleading. Comparability is of utmost importance in descriptive epidemiology of amputations, see in Kolossváry E, Ferenci T, Kováts T. Potentials, challenges, and limitations of the analysis of administrative data on vascular limb amputations in health care. Vasa. 2020 Mar;49(2):87-97.

Comment (8) Additionally to the preference for incidence rates (see above), I would suggest a major revision in results, avoiding confusion due to the numerosity of data (in text and tables). The authors provide data in each year by stratification according to age, sex, overall/major/minor procedures, PAD/AET. Instead of this, I would recommend using more graphics.

Line 181-188

„Women with PAD had up to 7.13% higher rate of major amputations compared to men, and women with AET of lower extremity arteries had an even greater disparity, with up to 10.02% higher rate compared to men (Table 2). The risk of major amputations was higher in women with PAD compared to men across all study years: 1.084 in 2015, 1.076 in 2016, 1.068 in 2017, 1.066 in 2018, and 1.059 185 in 2019. Similarly, women with AET of lower extremity arteries faced a higher risk of major amputations compared to men in most years: 1.271 in 2015, 1.123 in 2017, 1.057 in 2018, 187 and 1.046 in 2019. In 2016, the risk of major amputations was slightly higher in men compared to women (0.984).”

Comment (9) Please revise this part. In Table 2, in every category, I see a male predominance. Again, incidence providing is the appropriate approach to compare. No indication of how the authors calculated risk.

Line 237-257, Figure 8, 9. Comment (10) I can not see any advantage in age stratification (by decades). It is arbitrary; no similar data will be found for comparison (e.g. amputation below 30???). It is easy to illustrate age dependency by providing incidences on the Y axis and age on the X axis. The association is well-known in amputation literature but has not been forgotten again; the absolute numbers are misleading. The age structure of the background population is decisive. In Table 3-5 I can see incidences in different age strata; however, please clarify whether the base of this calculation is the whole population in a given age group and not the whole population general.

Discussion

Comment (11) Diabetes is considered as one of the main drivers of amputation practice. It would have been essential to address this question in the discussion. No data are available from Romania?

Comment (12) The minor to major amputations showed an increasing tendency. I would recommend offering an opinion about this. The increase in the incidence of minor amputations relative to major amputations has been observed in European countries and other populations, perhaps reflecting a shift in clinical practice toward limb-preserving amputations.

Line 291-318

Comment (13) This is a hot issue everywhere. The link between amputations and vascular procedures is not straightforward. Expecting a simple inverse relation seems misleading. Amputation practice is multifactorial. I am wondering whether the authors would be able to calculate the primary amputation rates in Romania? As a matter of fact, this segment can be substantial. Depending on case-mix, a higher rate of primary amputations may be acceptable (advanced cases, worse prognosis). Unfortunately, the present paper does not include an analysis of vascular care. Additionally, beyond the healthcare domain, socio-economic background is also influential. I would recommend broadening the focus in these directions.

Author Response

Please find the response for your comments in the attached file

Reviewer 2 Report

Comments and Suggestions for Authors

Comments

Five-years trends of vascular disease-related amputations in Romania: a retrospective database study

The authors assessed the rates of lower extremity amputations to peripheral artery disease and arterial embolism and thrombosis by age, sex, and amputation level in Romania during 2015-2019 timeframe. The authors refer to the substantial disparity in the burden of cardiovascular diseases between Western European countries and Central and Eastern European countries. The authors retrospectively analyzed data on PAD-and AET-related LEAs performed during admission to public hospitals in Romania between 1 January 2015 and 31 December 2019. Relevant data were selected from the national database.

The authors presented of LEAs per 100,000 population by province in Romania.  Here it would have been worthwile to summerize the importance and distribution of territorial inequality in text. The authors showed absolut number, crude incidence, and level amputation by year, incidence of vascular disease to lower extremity amputations/100,000 person/year, according to age in the general population (both PAD and AET also separately too). All this is contained in five tables. A total of nine figures present of results, they are well edited, easy to interpret and clear.  The textual interpretation of the figures is also correct.

The discussion contain the comparison of two Western European countries. A comparison with data of a Central and Eastern European country would also have been interesting. The absence of this does not detract from the value of the research.

The authors fully record the limitations of the study.

Although this study did not aim to present preventive strategies in lowering the LEAs in PAD and AET patients, but the data support the need to raise awareness this problem.

The study contains 21 references, the references are correct.

The purpose of the study, the source of data, the methodology of the analysis, the coherence of the results and discussion are realized, the conclusions drawn are correct.

Author Response

Please find the response to your comments in the attached file

Reviewer 3 Report

Comments and Suggestions for Authors

The study titled "Five-year trends of vascular disease-related amputations in Romania: a retrospective database study" presents statistics about amputations in Romania. The research is interesting to the readers and adds new information to the literature. The title is expressive about the contents of the study.

Abstract:

Well-written, while the conclusion is not related to the study data.

Introduction:

Objectives are clear and the knowledge gap is well stated.

Materials and Methods:

Why did you choose this time frame?

Please add the study definitions.

How did you deal with missing data?

Results:

Data are well presented.

Conclusion: this should be the take-home message and related to the presented data. 

Comments on the Quality of English Language

Accepted

Author Response

(The authors gave the same response as above.)

Round 2

Reviewer 1 Report

Comments and Suggestions for Authors

I appreciate the the authors honest effort to improve the scientific quality of paper. My remaining comments are as follows.

I will accept the author's preference for absolute numbers of procedures, although, in my view, they represent low value for readers. One thing is very important to understand. Future readers of the results will search for comparable data, and in this case, only incidence data are helpful.

In addition to providing sex and age group-specific incidence data, please incorporate the incidence values of all amputations, minor and major (general population in base). Yearly and average values are expected. These are the representatives of standard reporting.  

Consequently, I recommend putting these data in the abstract. As for the present form of the abstract, no one will be interested in how much different incidences were higher in men compared to women (Line 39 to41). Similarly, it seems not reasonable to put here the extent of the yearly increase in incidences with ps (Line 42 to 43). It is easier to say a significant increase in the period….

Author Response

Thank you for your valuable comments. in the attached file, please find our answer to your comments

Reviewer 3 Report

Comments and Suggestions for Authors

The authors responded to the previous comments

Comments on the Quality of English Language

accepted

Author Response

The answer to your comments are already uploaded on the previous version of the manuscript